# Recycled Sericin Hydrolysates Modified by Alcalase^®^ Suppress Melanogenesis in Human Melanin-Producing Cells via Modulating MITF

**DOI:** 10.3390/ijms23073925

**Published:** 2022-04-01

**Authors:** Keerati Joyjamras, Ponsawan Netcharoensirisuk, Sittiruk Roytrakul, Pithi Chanvorachote, Chatchai Chaotham

**Affiliations:** 1Graduate Program of Pharmaceutical Sciences and Technology, Faculty of Pharmaceutical Sciences, Chulalongkorn University, Bangkok 10300, Thailand; 6176453033@student.chula.ac.th; 2Department of Biochemistry and Microbiology, Faculty of Pharmaceutical Sciences, Chulalongkorn University, Bangkok 10330, Thailand; ponsawan.n@chula.ac.th; 3Functional Ingredients and Food Innovation Research Group, National Center for Genetic Engineering and Biotechnology, National Science and Technology Development Agency, Pathum Thani 12120, Thailand; sittiruk@biotec.or.th; 4Department of Pharmacology and Physiology, Faculty of Pharmaceutical Sciences, Chulalongkorn University, Bangkok 10330, Thailand; 5Center of Excellence in Cancer Cell and Molecular Biology, Faculty of Pharmaceutical Sciences, Chulalongkorn University, Bangkok 10330, Thailand

**Keywords:** wastewater, Alcalase^®^, hyperpigmentation, tyrosinase, human melanocyte, peptidomics

## Abstract

Because available depigmenting agents exhibit short efficacy and serious side effects, sericin, a waste protein from the silk industry, was hydrolyzed using Alcalase^®^ to evaluate its anti-melanogenic activity in human melanin-producing cells. Sericin hydrolysates consisted of sericin-related peptides in differing amounts and smaller sizes compared with unhydrolyzed sericin, as respectively demonstrated by peptidomic and SDS-PAGE analysis. The lower half-maximum inhibitory concentration (9.05 ± 0.66 mg/mL) compared with unhydrolyzed sericin indicated a potent effect of sericin hydrolysates on the diminution of melanin content in human melanoma MNT1 cells. Not only inhibiting enzymatic activity but also a downregulated expression level of tyrosinase was evident in MNT1 cells incubated with 20 mg/mL sericin hydrolysates. Quantitative RT-PCR revealed the decreased mRNA level of microphthalmia-associated transcription factor (MITF), a tyrosinase transcription factor, which correlated with the reduction of pCREB/CREB, an upstream cascade, as assessed by Western blot analysis in MNT1 cells cultured with 20 mg/mL sericin hydrolysates for 12 h. Interestingly, treatment with sericin hydrolysates for 6–24 h also upregulated pERK, a molecule that triggers MITF degradation, in human melanin-producing cells. These results warrant the recycling of wastewater from the silk industry for further development as a safe and effective treatment of hyperpigmentation disorders.

## 1. Introduction

Hyperpigmentation is the overproduction of melanin, which is a cellular pigment generated by melanocytes [1,2]. Although not being a communicable or lethal disease, hyperpigmentation disorders considerably reduce self-confidence as well as quality of life [1]. A large number of whitening cosmetic products have been developed to address the health and aesthetic concerns of hyperpigmented disorders [3]. Though various depigmenting agents such as kojic acid, arbutin, and glycolic acid have been introduced for hyperpigmentation treatment instead of hydroquinone, a first line therapy which exhibits blue-black pigmentation after a long-term application, most of them possess a short span of efficacy and skin irritation [4,5]. In order to manage this condition, laser and light sources have also been proposed with a good aesthetic outcome, but with a greater risk of side effects [6,7]. Therefore, natural extracts, which are an alternative resource of potentially active compounds, have gained attention as safe and effective hypopigmentation agents [8].

Melanocytes laid at the deepest layer of epidermis are responsible for melanin production in the melanogenesis pathway [9]. Although composed of various regulatory proteins, tyrosinase is considered as the rate-limiting enzyme that catalyzes the conversion of tyrosine to melanin. Thus, tyrosinase inhibitors have been widely accepted as effective depigmenting agents for cosmeceutical application [10]. The transcription of tyrosinase mRNA is stimulated by microphthalmia-associated transcription factor (MITF) [11]. Cyclic adenosine 3′,5′-monophosphate (cAMP) also mediates melanin production. The activation of cAMP further phosphorylates cAMP response element-binding protein (CREB), which provokes MITF transcription [12]. Additionally, β-catenin mediated by glycogen synthase kinase 3β (GSK3β) also modulates MITF expression at the transcription level [13]. It is worth noting that MITF is post-translationally regulated by phosphorylated extracellular signal-regulated kinase (pERK) through proteasomal degradation, which leads to a diminished tyrosinase level and suppression of melanin synthesis [14,15]. According to the regulatory role on tyrosinase expression, MITF has been proposed as a novel target for anti-melanogenic compounds that may provide a sustainable effect on inhibiting melanin production [8,16].

Recently, the biological functions of sericin, a waste protein presented in the degumming water during silk processing, were discovered [17,18,19]. Moreover, sericin hydrolysates obtained from the enzymatic modification by Alcalase^®^ possess biological activities greater than the native form [20]. Though the inhibitory potential of sericin protein against the enzymatic activity of mushroom tyrosinase has been shown [21,22], the anti-melanogenesis effects of sericin hydrolysates prepared by Alcalase^®^ in human melanin-producing cells have not been investigated. Thus, the inhibitory effect on melanin production and related mechanisms of sericin hydrolysates prepared by Alcalase^®^ in human melanoma MNT1 cells were evaluated in this study. The obtained information would warrant the recycling and utilization of sericin, a useless waste product released from the silk industry, as a potent hypopigmentation agent for further development of safe and effective cosmeceutical products.

## 2. Results

### 2.1. Peptide Constituents in Sericin Hydrolysates Modified by Alcalase^®^

The alteration of peptide components contained in Alcalase^®^-hydrolyzed sericin was initially evaluated by SDS-PAGE. As shown in Figure 1a, SDS-PAGE with Coomassie brilliant blue R-250 clearly showed the disappearance of large molecular weight peptides (~15–260 kDa) in the sericin hydrolysates compared with unhydrolyzed sericin. Notably, Alcalase^®^-hydrolyzed sericin contained primarily low molecular weight peptides at ~10 kDa. The modification of peptide constituents after enzymatic reaction with Alcalase^®^ was further evaluated by LC-ESI-MS/MS compared with native peptides present in unhydrolyzed sericin. The data obtained from peptidomic analysis indicate the presence of various sericin-related peptides in the unhydrolyzed sericin, which are proteins made from the wastewater of the silk industry (Figure 1b). Though mostly composed of sericin-derived peptides, the amount of peptide constituents was remarkably different in the sericin hydrolysates compared with unhydrolyzed sericin. These results support the modification of peptide constituents in sericin hydrolysates prepared by Alcalase^®^.

### 2.2. Suppressive Effect of Sericin Hydrolysates on Melanin Production in Human Melanin-Producing Cells

The cytotoxic profile of Alcalase^®^-hydrolyzed sericin was accessed using MTT viability assay before evaluating the suppressive effect on melanin production in human melanin-producing cells. After culturing for 24 h, Figure 2a demonstrates no significant change in MNT1 cell viability following treatment with 1–20 mg/mL sericin hydrolysates compared with the untreated cells. The effect on cell proliferation was further examined in MNT1 cells incubated for 24–72 h with Alcalase^®^-hydrolyzed sericin at 0–20 mg/mL. Despite a 24 h treatment with a considerably non-toxic concentration, sericin hydrolysates (20 mg/mL) notably inhibited proliferation in human MNT1 cells over a 48–72 h incubation (Figure 2b). Thus, treatment with Alcalase^®^-hydrolyzed sericin ranging from 0 to 20 mg/mL for 0–24 h was chosen as the non-toxic condition for determining anti-melanogenic activity in human melanin-producing cells.

Figure 2c shows the gradual increase of melanin content in MNT1 cells after culturing for 6–24 h. The cAMP activator and tyrosinase inhibitor, forskolin and 4-butylresorcinol, were respectively used as positive and negative controls [23,24]. Surprisingly, incubation for 12 h with either sericin hydrolysates (20 mg/mL) or 4-butylresorcinol (20 μM) comparably decreased cellular melanin content in human melanin-producing cells (Figure 2d). It is worth noting that the reduction of melanin content in a dose-dependent manner was observed in MNT1 cells after a 24 h treatment with sericin hydrolysates at 5–20 mg/mL. The inhibitory effect on melanin production of unhydrolyzed sericin was also evaluated in human melanoma MNT1 cells. Compared with sericin hydrolysates modified by Alcalase^®^ at the same concentration (1–20 mg/mL), a higher amount of cellular melanin was presented in MNT1 cells cultured with unmodified sericin for 24 h (Figure 3a). This corresponded with the half-maximum inhibitory concentration (IC_50_) on melanin production in MNT1 cells shown in Figure 3b. The lower IC_50_ value was demonstrated in sericin hydrolysates at approximately 9.05 ± 0.66 mg/mL compared with an IC_50_ value of 24.54 ± 0.17 mg/mL for the unhydrolyzed sericin-treated group. Taken together, these results clearly demonstrate the potent inhibitory effect of sericin hydrolysates against melanin production in human melanin-producing cells.

### 2.3. Alcalase^®^-Hydrolyzed Sericin as a Human Tyrosinase Inhibitor

The role as human tyrosinase inhibitor was determined after adding l-DOPA into the cellular lysate prepared from MNT1 cells that contains tyrosinase with or without Alcalase^®^-hydrolyzed sericin. The formed dopachrome, which represents tyrosinase activity was reduced in a concentration-dependent manner in the reaction containing 1–20 mg/mL sericin hydrolysates (Figure 4). It was worth noting that the significant inhibition of human tyrosinase activity was observed only after the incubation with 10–20 mg/mL of sericin hydrolysates. Nevertheless, the results confirm a role of Alcalase^®^-hydrolyzed sericin as an inhibitor against enzymatic activity of human tyrosinase.

### 2.4. Sericin Hydrolysates Downregulate Tyrosinase Expression in Human Melanin-Producing Cells

The expression level of tyrosinase, the rate-limiting enzyme in melanogenesis, was further evaluated to clarify the underlying mechanisms of Alcalase^®^-hydrolyzed sericin in human melanin-producing cells. Quantitative reverse transcription PCR revealed that both mRNA levels of tyrosinase and MITF, a tyrosinase transcription factor, were downregulated in MNT1 cells cultured with 20 mg/mL sericin hydrolysates for 12 h (Figure 5a).

Consequently, the decreased levels of both MITF (Figure 5b) and tyrosinase proteins (Figure 5c) were detected by Western blot analysis in human melanin-producing cells incubated with sericin hydrolysates (20 mg/mL) for 24 h. It is worth noting that the modulation of MITF and tyrosinase expression levels is well-correlated with the anti-melanogenic activity of Alcalase^®^-hydrolyzed sericin observed in MNT1 cells after a 24 h treatment.

### 2.5. Alteration of MITF-Regulating Proteins in Human Melanin-Producing Cells Cultured with Sericin Hydrolysates

Based on the decreased expression of MITF mRNA, the modulation of upstream molecules regulating MITF transcription was examined in sericin hydrolysates-treated MNT1 cells. Despite no alteration of pGSK3β/GSK3β (Figure 6a) and β-catenin protein levels (Figure 6b), there was a significant reduction of pCREB/CREB expression in MNT1 cells incubated with 20 mg/mL sericin hydrolysates for 12 h compared with the control cells (Figure 6c). The suppression of the pCREB/CREB cascade was correlated with a reduced MITF mRNA level in MNT1 cells following treatment with sericin hydrolysates for 12 h. Because of the remarked decrease of MITF protein expression at 24 h, the role of Alcalase^®^-hydrolyzed sericin on regulatory proteins involved in the post-translational modification of MITF were examined. Figure 6d shows the upregulation of pERK, a signaling molecule that promotes MITF degradation, in human melanin-producing cells incubated with 20 mg/mL sericin hydrolysates for 6–24 h. These results suggest that Alcalase^®^-hydrolyzed sericin mediates MITF expression at both the transcription and post-translational level.

## 3. Discussion

In searching for an effective treatment for hyperpigmented disorders, it is essential to identify compounds that not only exhibit potent efficacy but also have an acceptable safety profile [4,25]. Recent evidence indicates that MITF, a melanogenic transcription factor, is a novel therapeutic target for modulating melanin synthesis [8,16]. Suppression of MITF correlates with downregulated melanogenesis, reduction of melanin content, and lightened skin tone [16]. While most of the available depigmenting agents on the market are tyrosinase inhibitors [10,26], sericin hydrolysates prepared by using Alcalase^®^ exhibit potent anti-melanogenic activity in human melanin-producing cells by acting as human tyrosinase inhibitor (Figure 4) and modulating MITF expression (Figure 5a,b).

Sericin, a natural protein present in the degumming water from the silk industry, is recognized for its therapeutic benefits, especially in cosmeceutical applications [27]. Nevertheless, a broad peptide composition (~10–250 kDa) hinders sericin from precise therapeutic targeting and lessens its potency [28,29,30]. Optimal size distribution and specific sequences of amino acids influence the biological activity of natural proteins [31]. Because of the adjustable conditions for producing protein hydrolysates, enzymatic reactions are widely accepted as a method to modify protein structure and composition [32,33]. Among the various modifying proteases, Alcalase^®^-hydrolyzed proteins exhibited the most promising therapeutic effects [20]. As such, the altered pattern of both size (Figure 1a) and peptide component (Figure 1b) showed a higher anti-melanogenic activity of Alcalase^®^-hydrolyzed sericin compared with the unmodified sericin protein (Figure 3). Accounting for a wide range of substrate specificity, availability as a commercial enzyme [34,35], and potent anti-melanogenic effects demonstrated in this study, Alcalase^®^ could be considerably suitable for enzymatic modification of sericin protein.

The greater activity of sericin hydrolysates could result from smaller-sized peptides, which readily permeate through cell membranes [36]. Despite the diverse types of sericin-related peptides, specific peptides identified by peptidomic analysis in Alcalase^®^-hydrolyzed sericin likely play a crucial role in the inhibition of melanin production in human melanin-producing cells. Amino acid constituents in peptides critically modulate the inhibitory activity against enzymatic function of tyrosinase. It has been reported that tyrosinase inhibitory peptides should be composed with arginine and/or phenylalanine for strong binding with tyrosinase and inhibitory activity. Additionally, hydrophobic amino acids such as valine, alanine, and leucine, as well as aromatic residual of threonine and tyrosine, are also essential for the activity of tyrosinase inhibitory peptides [37,38]. The anti-melanogenic potential of sericin was discovered through its activity as a mushroom tyrosinase inhibitor [21,22]. Besides being composed of various inhibitory amino acids, serine and threonine contained in sericin was proposed to chelate copper, which involves the capability of melanin synthesis at the active site of mushroom tyrosinase [39]. Though mushroom tyrosinase is wildly accepted for screening of tyrosinase inhibitors [40,41], the difference of structure and amino acid sequence between human and mushroom tyrosinase could result in variable effects of the candidate tyrosinase inhibitors [42,43]. The inhibitory activity against human tyrosinase associated with sericin-derived peptides (Figure 4), which was first identified in this study, warrants the further development of sericin hydrolysates prepared by using Alcalase^®^ as an effective treatment for hyperpigmented disorders.

Interestingly, the suppression on both activity (Figure 4) and protein expression of tyrosinase (Figure 5a,c), the rate-limiting enzyme in melanogenesis, indicates the inhibitory effect of Alcalase^®^-hydrolyzed sericin against melanin production observed in human melanin-producing cells after a 12 and 24 h treatment (Figure 2c,d). It should be noted that the modulated mRNA level should be detected prior to the alteration of protein expression. A significant reduction of tyrosinase mRNA and its transcription factor, MITF, was demonstrated in MNT1 cells cultured with 20 mg/mL sericin hydrolysates at 12 h (Figure 5a). This was correlated with the concomitant reduction of protein levels at a later time point (Figure 5b,c).

The upregulated proteins in the MITF-related pathways result in the overproduction of melanin and cell hyperproliferation [44]. Consequently, depigmenting compounds that target MITF effectively reduce melanin content via restraining melanogenesis-regulating proteins and inhibiting the proliferation of melanocytes [16,45]. Notably, MNT1 cells are human melanoma and have been used for evaluating melanogenic effects because of their hyperproliferation and hyperpigmented phenotypes [46,47]. The low proliferation (Figure 2b) as well as decreased expression of MITF and tyrosinase in MNT1 cells incubated with sericin hydrolysates (Figure 5) demonstrate the high efficacy of Alcalase^®^-hydrolyzed sericin against melanogenesis in human melanin-producing cells. Nevertheless, the anti-melanogenic effect of Alcalase^®^-hydrolyzed sericin should be further investigated in animal and clinical studies.

The modulatory role of sericin hydrolysates prepared using Alcalase^®^ on the upstream signaling cascade that regulates MITF was observed by the reduction of the pCREB/CREB level (Figure 6c); however, there was no alteration in the GSK3β/β-catenin signal (Figure 6a,b). The suppression of pCREB/CREB signaling correlated with a decreased MITF mRNA level in human melanin-producing cells after treatment with sericin hydrolysates for 12 h. It should be noted that the gradual increase in pCREB/CREB expression level after a 24 h treatment with sericin hydrolysates might be the result of a positive feedback mechanism from downstream proteins [48]. Despite the recovery of the pCREB/CREB signal at the later time point, the low expression of MITF mRNA and MITF protein was sustained for up to 24 h of treatment with Alcalase^®^-hydrolyzed sericin. The protein level of MITF is post-translationally regulated by pERK through proteasomal degradation [49]. The upregulated pERK/ERK pathway in human melanin-generating cells treated with sericin hydrolysates (Figure 6d) would promote the reduction of MITF protein concomitantly with downregulated tyrosinase and a reduction of melanin production.

## 4. Materials and Methods

### 4.1. Chemical Reagents

Sodium dodecyl sulfate (SDS), radio-immunoprecipitation assay (RIPA) lysis buffer, Coomassie brilliant blue R-250, isopropanol, ethanol, methanol, acetic acid solution, ammonium bicarbonate, formic acid, acetonitrile, sodium hydroxide solution (NaOH), synthetic melanin, bovine serum albumin (BSA), chloroform, 2-propanol, 38% (*w/v*) formaldehyde solution, dimethyl sulfoxide (DMSO), crystal violet solution, forskolin and 4-butylresorcinol were bought from Sigma-Aldrich (St. Louis, MO, USA). Luna^®^ Universal qPCR Master Mix, RevertAid First Stand cDNA synthesis kit, DNase I, Bicinchoninic acid (BCA) protein assay kit, and 3-(4,5-dimethyl-2-thiazolyl)-2,5-diphenyl-2*H*-tetrazolium bromide (MTT) were procured from Thermo Scientific (Rockford, IL, USA). *N*,*N*,*N*′,*N*′-tetramethylethylenediamine (TEMED), 40% acrylamide/bisacrylamide 37.5:1, Tween 20, and ammonium persulfate were obtained from Bio-Rad (Hercules, CA, USA). GENEzol reagent was bought from Geneaid Biotech Ltd. (New Taipei City, Taiwan). Santa Cruz Technology (Dallas, TX, USA) and Abcam (Cambridge, UK) were a source of primary antibody to tyrosinase and MITF, respectively. Meanwhile, primary antibody to CREB, pCREB (Ser133), ERK1/2, pERK1/2 (Thr202/Tyr204), GSK3β, pGSK3β (Ser9), β-catenin, and GAPDH as well as horseradish peroxidase (HRP)-linked specific secondary antibodies were obtained from Cell Signaling Technology (Denver, MA, USA).

### 4.2. Preparation of Sericin Hydrolysates

Enzymatic modification of sericin was performed using Alcalase^®^ (EC 3.4.21.62; Sigma-Aldrich, St. Louis, MO, USA) as previously reported [20]. Briefly, lyophilized sericin powder, a generous gift from Ruenmai-baimon, Ltd., Surin Province, Thailand, was dispersed in de-ionized water and further heated at 95 °C for complete solubilization for 10 min. The solution was put on ice for cooling down to room temperature. The hydrolysis by Alcalase^®^ was performed at enzyme/substrate ratio: 2, pH 8 at 60 °C for 3 h as recommended by the manufacturer’s instructions. Then, the reaction was heated until 100 °C for 5 min to stop enzymatic function of Alcalase^®^ followed with being immediately chilled on ice. Then, the supernatant was collected through 3500× *g* (4 °C) for 30 min and further dialyzed overnight with the phosphate buffer at 4 °C. The obtained solution was then freeze-dried to collect the powder of sericin hydrolysates that were kept at −20 °C until use.

### 4.3. Molecular Weight Distribution of Sericin Hydrolysates

Molecular weight distribution of sericin hydrolysates was determined by sodium dodecyl sulfate polyacrylamide gel electrophoresis (SDS-PAGE) analysis. The protein samples approximately at 30 µg were separated using 12% (*w/v*) gel of SDS-PAGE. The separated protein constituents were then stained with Coomassie brilliant blue R-250 solution overnight. The excess dye was removed by soaking the gel with destain solution composed of isopropanol:acetic acid:water at 10:10:80% *v/v* [50].

### 4.4. Peptidomic Analysis of Alcalase^®^-Hydrolyzed Sericin

#### 4.4.1. Liquid Chromatography/Electrospray Ionization–Tandem Mass Spectroscopy (LC-ESI-MS/MS)

Both sericin hydrolysates and unhydrolyzed sericin was solubilized in 10 mM ammonium bicarbonate. Before being subjected to LC-ESI-MS/MS analysis, the zip-tip purified protein samples were dried using a speed vacuum concentrator (Thermo Scientific, Loughborough, UK) and further dispersed in 0.1% formic acid. The protein solutions were injected into an Ultimate3000 Nano/Capillary LC System (Thermo Scientific, Loughborough, UK) coupled to a Hybrid quadrupole Q-Tof impact II™ (Bruker Daltonics GmbH, Bremen, Germany) equipped with a Nano-captive spray ion source. Briefly, 1 µL of peptide digests were enriched on a µ-Precolumn 300 µm i.d. × 5 mm C18 PepMap 100, 5 µm, 100 A (Thermo Scientific, Loughborough, UK), separated on a 75 μm i.d. × 15 cm and packed with Acclaim PepMap RSLC C18, 2 μm, 100 Å, nanoViper (Thermo Scientific, Loughborough, UK). The C18 column was enclosed in a thermostatted column oven set to 60 °C. Solvent A and B containing 0.1% formic acid in water and 0.1% formic acid in 80% acetonitrile, respectively, were supplied on the analytical column. A gradient of 5–55% solvent B was used to elute the peptides at a constant flow rate of 300 nL/min for 30 min. Electrospray ionization was carried out at 1.6 kV using the CaptiveSpray. Nitrogen was used as a drying gas (flow rate about 50 Lh). Collision-induced dissociation (CID) product ion mass spectra were obtained using nitrogen gas as the collision gas. Mass spectra (MS) and MS/MS spectra were obtained in the positive-ion mode at 2 Hz over the range (*m/z*) 150–2200. The collision energy was adjusted to 10 eV as a function of the *m/z* value. Triplicates of LC-MS analysis were performed in each sample.

#### 4.4.2. Bioinformatics and Data Analysis

The amount of peptide constituents was quantified by MaxQuant 1.6.6.0 (Max Planck Institute for Biochemistry, Planegg, Germany) using the Andromeda search engine to correlate MS/MS spectra to the UniProt sericin database [51]. Label-free quantitation with MaxQuant’s standard settings was performed: maximum of two miss cleavages, mass tolerance of 0.6 dalton for main search, unspecific digesting enzyme, oxidation of methionine as variable modifications. The identification of protein was proceeded if there was one unique peptide with at least 7 amino acids. For further data analysis, the identified protein required two peptides with one unique peptide. Protein FDR was set at 1% and estimated using the reversed search sequences. The maximal number of modifications per peptide was set to 5. As a search FASTA file, the proteins present in the sericin proteins were downloaded from UniProt. Potential contaminants present in the contaminants.fasta file that comes with MaxQuant were automatically added to the search space by the software. The levels of peptide constituents were presented as log2 value.

### 4.5. Cytotoxicity of Sericin Hydrolysates in Human Melanin-Producing Cells

Human melanoma MNT1 cells (ATCC, Manassas, VA, USA) were maintained in Dulbecco’s modified Eagle medium (DMEM) with 20% (*v/v*) fetal bovine serum (FBS), 10% AIM-V medium, 2 mmol/L l-glutamine, and 100 units/mL of penicillin/streptomycin (Gibco, Gaithersburg, MD, USA). The cells were cultured in a humidified incubator at 37 °C with 5% CO_2_ until 70–80% confluence prior to being used in further experiments.

The cytotoxicity of sericin hydrolysates in human MNT1 cells was determined via MTT assay. Briefly, MNT1 cells (5 × 10^3^ cells/well in a 96-well plate) were treated with 0–20 mg/mL of sericin hydrolysates for 24 h. Then, the cells were further incubated with 0.5 mg/mL of MTT solution for 3 h at 37 °C in the dark. A microplate reader (PerkinElmer Inc., Waltham, MA, USA) was used for measurement of the optical density (OD) of formazan crystal solubilized in DMSO at 570 nm. A relative OD ratio between sericin hydrolysate-treated cells to the non-treated control group was calculated and presented as the percent (%) cell viability.

The effect of sericin hydrolysates on proliferation of human melanin-producing cells was also evaluated by crystal violet staining [52]. MNT1 cells seeded at 2 × 10^3^ cells/well in a 96-well plate were cultured with sericin hydrolysates (0–20 mg/mL) for 24–72 h. After removal of unattached dead cells by washing two times with 100 µL de-ionized water, 1% (*w/v*) formaldehyde was added to fix the attached living cells for 30 min. Then, 0.05% (*w/v*) crystal violet was added to stain the cells for another 30 min. The excess crystal violet was removed by washing twice with 100 µL de-ionized water and then the plate was air-dried overnight. The OD of cellular crystal violet dissolved in methanol was determined at 570 nm using a microplate reader. The OD of the treatment at each time point was relatively compared to the OD of non-treated control cells at 24 h and presented as the relative proliferation.

### 4.6. Measurement of Cellular Melanin Content

Melanin-producing MNT1 cells (1 × 10^5^ cells/well in 24-well plates) were treated with either sericin hydrolysates (1–20 mg/mL), 10 μM forskolin, or 20 μM 4-butylresorcinol for 6–24 h. After indicated time points, the cell membrane was broken by the incubation on ice for 45 min with RIPA buffer containing 1% protease inhibitor cocktail (Roche Molecular Biochemicals, Indianapolis, IN, USA). After centrifugation at 12,000 rpm × 10 min (4 °C), the separated melanin pigments were solubilized in 1N NaOH in 10% DMSO (200 μL) at 80 °C for 3 h. The melanin content was interpolated from the absorbance intensity at 405 nm of standard melanin. The amount of total protein determined from the BCA assay was used to calculate the melanin content/µg of protein.

### 4.7. Determination of Inhibitory Activity of Sericin Hydrolysates against Human Tyrosinase

The inhibitory effect of sericin hydrolysates on enzymatic activity of human tyrosinase was determined as per previous reports [53,54]. MNT1 cells at 70–80% confluence were lysed with RIPA buffer composed of 1% protease inhibitor at 4 °C and were subsequently vortexed every 10 min for 1 h. The supernatant separated by centrifugation at 12,000 rpm for 15 min (4 °C) was determined for protein content using BCA assay. The sample containing 50 µg of protein was incubated with different concentrations (0–20 mg/mL) of sericin hydrolysates at 37 °C for 10 min. Then, 2 mM l-DOPA, a substrate of tyrosinase, was added into the MNT1 cellular lysate and the reaction mixture was further incubated at 37 °C for 2 h. The absorbance of formed dopachrome was determined at 490 nm using a microplate reader. The tyrosinase activity was calculated as follows.
$$\mathrm{Tyrosinase}\;\mathrm{activity}\;(\%)=\frac{\mathrm{Absorbance}\;\mathrm{at}\;490\;\mathrm{nm}\;\mathrm{of}\;\mathrm{treatment}\;}{\mathrm{Absorbance}\;\mathrm{at}\;490\;\mathrm{nm}\;\mathrm{of}\;\mathrm{control}}\times \;100\%$$

### 4.8. Western Blot Analysis

After indicated treatment, MNT1 cells (1 × 10^5^ cells/well in a 6-well plate) were incubated in iced-cold RIPA lysis buffer with 1% protease inhibitor cocktail for 30 min. The cell lysate containing 30 µg protein assessed by BCA protein assay from each was mixed with loading dye before separating onto 10% SDS-PAGE. Thereafter, the separated proteins were transferred from the gel onto nitrocellulose membrane (Bio-Rad, Hercules, CA, USA). The membrane was immersed into 5% non-fat skim milk in TBST (25 mM Tris-HCl; pH 7.5, 125 mM NaCl, 0.1% Tween20) for 1 h at room temperature. Then, the membrane was washed with TBST for 5 min prior to incubation with primary antibody at 4 °C for 12 h. Before probing with HRP-secondary antibody for 2 h at room temperature, the membrane was washed three times (5 min) with TBST. The band of interested protein was examined after adding of chemiluminescence substrate (SuperSignal West Pico, Pierce, Rockford, IL, USA) onto the membrane. The signal of interested protein band was visualized and determined using Chemiluminescent ImageQuant LAS 4000 (GE Healthcare Bio-Sciences AB, Björkgatan, Uppsala, Sweden).

### 4.9. Quantitative Reverse Transcription Polymerase Chain Reaction (RT-qPCR)

Total RNA from MNT1 cells (1 × 10^5^ cells/well in a 6-well plate) treated with 0–20 mg/mL sericin hydrolysates for 6–24 h were extracted using GENEzol^TM^ reagent and used as a template for synthesis cDNA using RevertAid Premium Reverse Transcriptase. The cDNA was amplified by specific forward and reverse primers (Thermo Fisher Scientific, Waltham, MA, USA) as follows.

-Human MITF forward primer: 5′-TCATCCAAAGATCTGGGCTATGACT-3′-Human MITF reverse primer: 5′-GTGACGACACAGCAAGCTCAC-3′-Human tyrosinase forward primer: 5′-TCATCCAAAGATCTGGGCTATGACT-3′-Human tyrosinase reverse primer: 5′-GTGACGACACAGCAAGCTCAC-3′-Human GAPDH forward primer: 5′-GAGTCCACTGGCGTCTTCA-3′-Human GAPDH reverses primer: 5′-TTCAGCTCAGGGATGACCTT-3′

RT-qPCR was performed through a C1000™ Thermal Cycler (Bio-Rad CFX384 real-time PCR system) using Luna^®^ Universal qPCR Master Mix. The reaction combinations were incubated for 39 cycles of 95 °C for 5 s, 60 °C for 30 s, and 65 °C for 5 s. The comparative threshold (Ct) of target genes was normalized to the GAPDH values. The relative comparison by the ΔCt method was established to determine the expression of target mRNA.

### 4.10. Statistical Analysis

All experimental data were presented as means ± standard error of the mean (SEM). SPSS version 22 (IBM Corp., Armonk, NY, USA) with one-way analysis of variance (ANOVA) followed by Tukey’s post-hoc test was performed for statistical analysis. Any *p*-value under 0.05 was considered statistically significant.

## 5. Conclusions

Sericin hydrolysates prepared by using Alcalase^®^ suppress melanin production in human melanin-producing cells by inhibiting tyrosinase activity and modulating MITF-related proteins (Figure 7). The regulation of MITF at both the transcription and post-translational levels mediated by Alcalase^®^-hydrolyzed sericin is associated with reduced pCREB and upregulated pERK levels, respectively. The concomitant reduction of MITF expression results in downregulated levels of tyrosinase and melanin content in human melanin-producing cells cultured with sericin hydrolysates. The results suggest the use of recycled waste products from the silk industry for further development as a potentially safe and effective treatment for hyperpigmentation disorders.

## Figures and Tables

**Figure 1 ijms-23-03925-f001:**
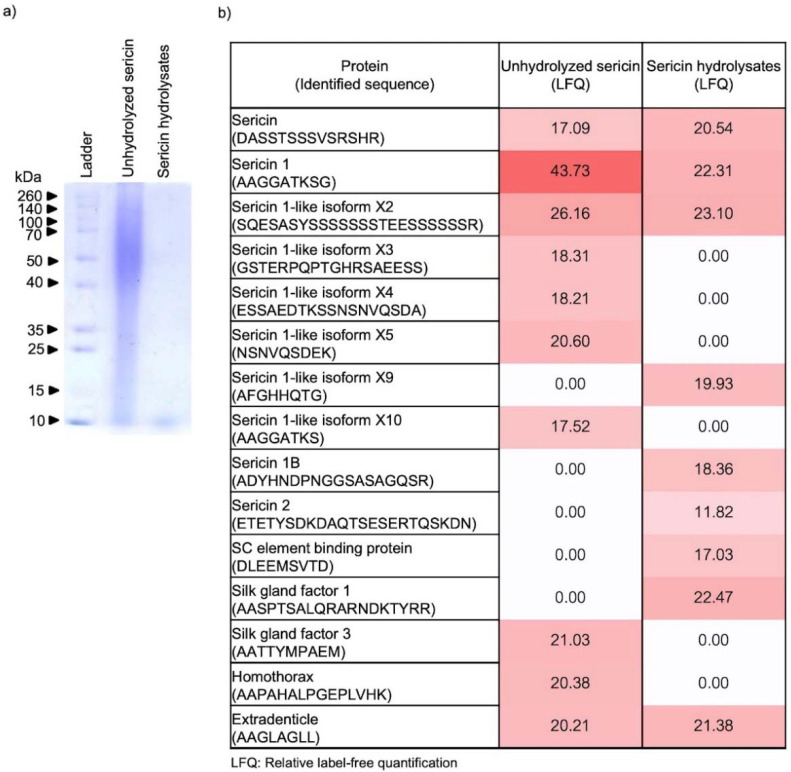
Peptide constituents in sericin hydrolysates modified by Alcalase^®^. (**a**) The wide range of peptide distribution from 10 to 260 kDa was demonstrated in unmodified sericin via SDS-PAGE analysis while only peptide at ~10 kDa was contained in sericin hydrolysates prepared using Alcalase^®^. (**b**) Peptidomic analysis revealed the alteration of both type and ratio of peptide constituents in sericin after enzymatic modification by Alcalase^®^. The level of peptides in each sample were expressed as log2 value.

**Figure 2 ijms-23-03925-f002:**
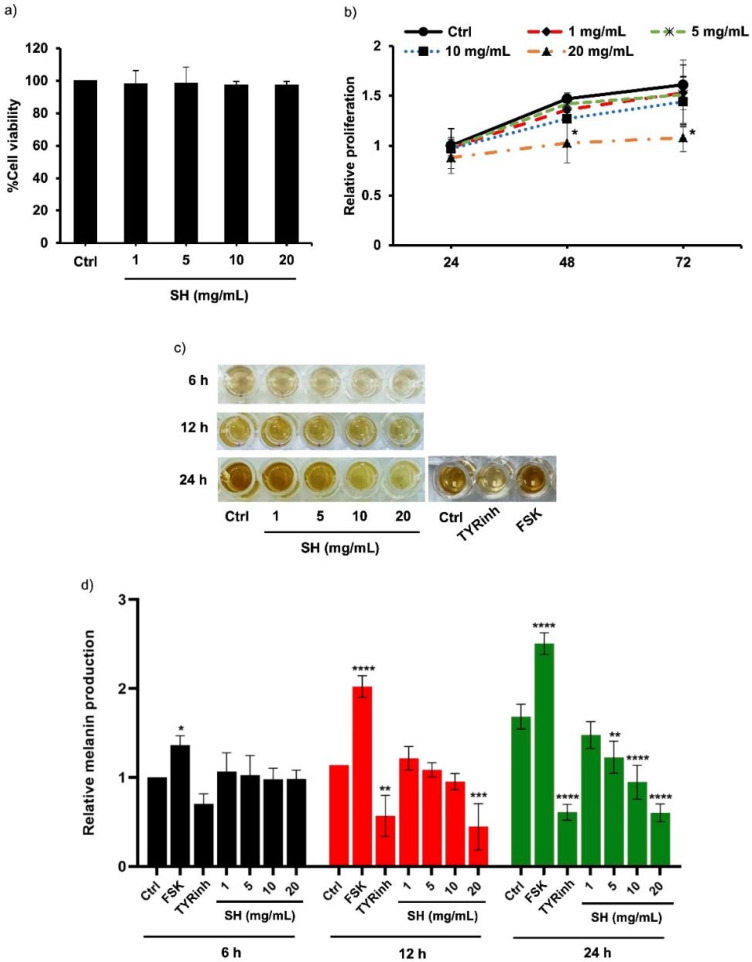
Inhibitory effect of Alcalase^®^-hydrolyzed sericin on melanin production in human melanin-producing cells. (**a**) Despite no alteration of %cell viability detected via MTT assay after 24 h treatment with 1–20 mg/mL sericin hydrolysates (SH), (**b**) crystal violet assay indicated low proliferation in human MNT1 cells cultured with 20 mg/mL sericin hydrolysates for 48–72 h. (**c**) The melanin content was gradually increased in MNT1 cells after culturing for 6–24 h in the experiment condition. Notably, the remarkable decrease and increase of cellular melanin was respectively observed in MNT1 cells treated with 20 μM tyrosinase inhibitor (TYRinh; 4-butylresorcinol) and 10 μM forskolin (FSK) for 24 h. (**d**) Interestingly, sericin hydrolysates at 20 mg/mL significantly decreased melanin content in human melanin-producing cells promptly at 12 h treatment and sustainably until 24 h of the incubation time. Data are presented as means ± SEM from three independent experiments. * *p* < 0.05, ** *p* < 0.01, *** *p* < 0.005, **** *p* < 0.001 versus non-treated cells (Ctrl).

**Figure 3 ijms-23-03925-f003:**
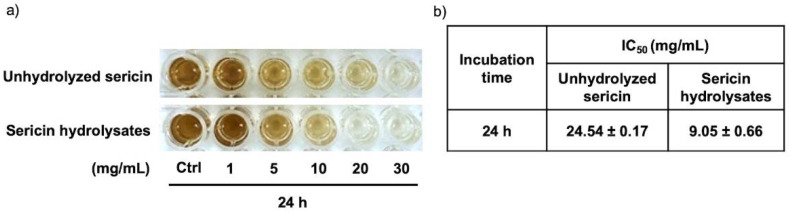
The more potency of Alcalase^®^-hydrolyzed sericin on suppression of melanin production when compared with unhydrolyzed sericin was evidenced by (**a**) the lower cellular melanin content and (**b**) half-maximum inhibitory concentration (IC_50_) in human melanin-producing cells after 24 h of treatments.

**Figure 4 ijms-23-03925-f004:**
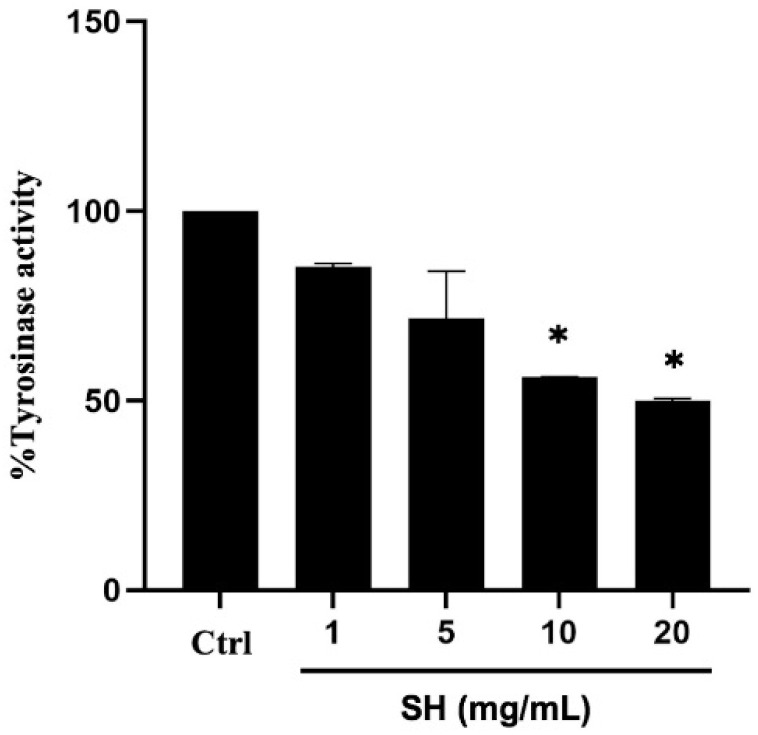
The enzymatic function of tyrosinase contained in the cellular lysate from human MNT1 cells was determined after the adding of its substrate, l-DOPA (2 mM) with or without sericin hydrolysates (SH) prepared using Alcalase^®^. After further incubation at 37 °C for 2 h, Alcalase^®^-hydrolyzed sericin at 10–20 mg/mL significantly inhibited enzymatic activity of human tyrosinase. Data are presented as means ± SEM from three independent experiments. * *p* < 0.05 versus non-treated cells (Ctrl).

**Figure 5 ijms-23-03925-f005:**
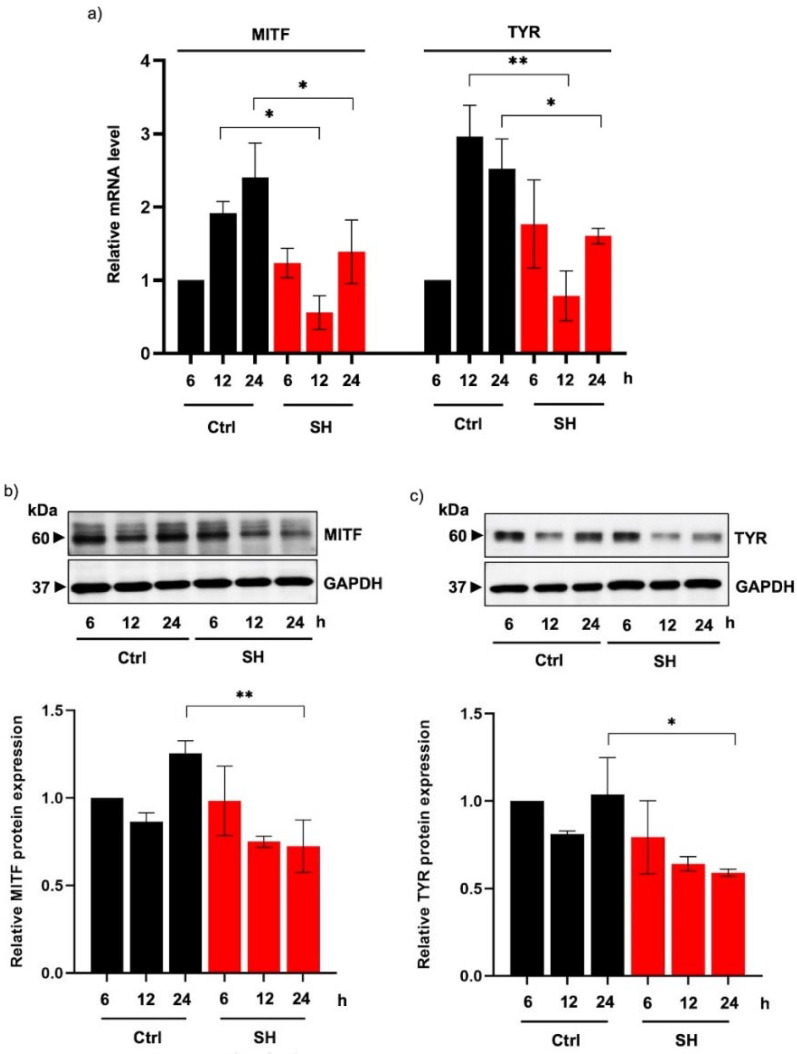
Diminution of tyrosinase expression in human melanin-generating cells cultured with sericin hydrolysates. (**a**) Quantitative real-time PCR demonstrated the decreased mRNA levels of MITF and tyrosinase (TYR) in human MNT1 cells incubated with 20 mg/mL sericin hydrolysates (SH) prepared by Alcalase^®^ for 12–24 h. Consequently, protein expression levels of (**b**) MITF and (**c**) tyrosinase was diminished in sericin hydrolysates-treated human melanin-producing cells after 24 h of the incubation time. Data are presented as means ± SEM from three independent experiments. * *p* < 0.05, ** *p* < 0.01 versus non-treated cells (Ctrl).

**Figure 6 ijms-23-03925-f006:**
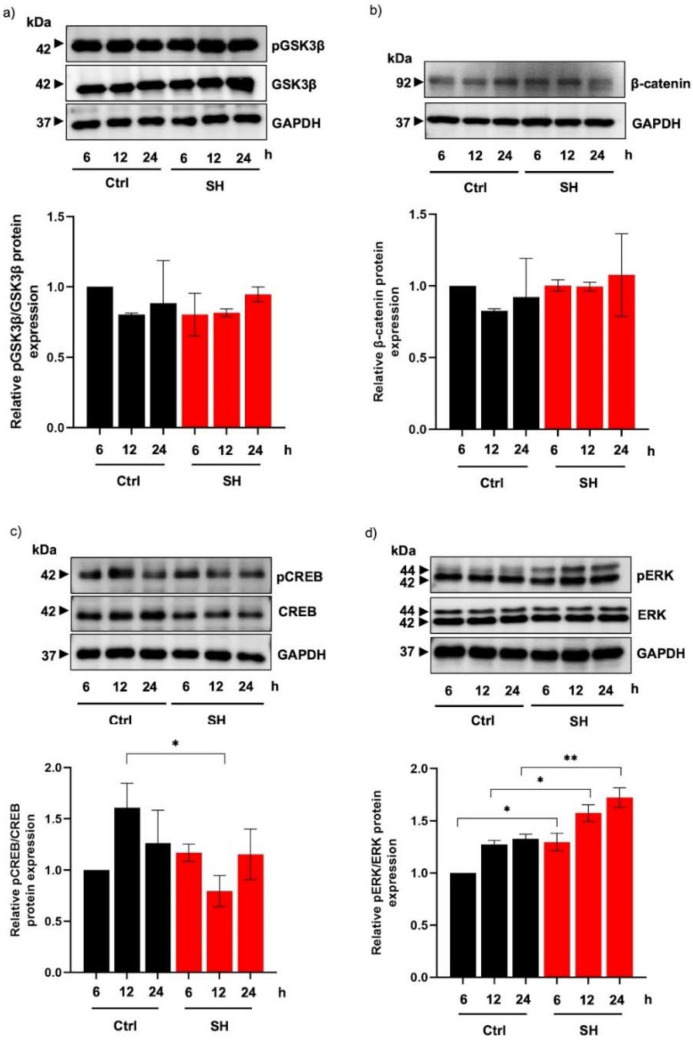
MITF-regulating proteins modulated by sericin hydrolysates prepared by Alcalase^®^ (**a**,**b**). Although there was no alteration of protein involving GSK3β/β-catenin cascade, (**c**) the decreased expression of pCREB/CREB indicated the restraint on upstream signal regulating MITF transcription in human MNT1 cells incubated with 20 mg/mL sericin hydrolysates (SH) for 12 h. (**d**) Additionally, treatment with Alcalase^®^-hydrolyzed sericin (20 mg/mL) upregulated the expression of pERK, a signaling protein stimulating MITF degradation, in human melanin-producing cells after the incubation for 6–24 h. Data are presented as means ± SEM from three independent experiments. * *p* < 0.05, ** *p* < 0.01 versus non-treated cells (Ctrl).

**Figure 7 ijms-23-03925-f007:**
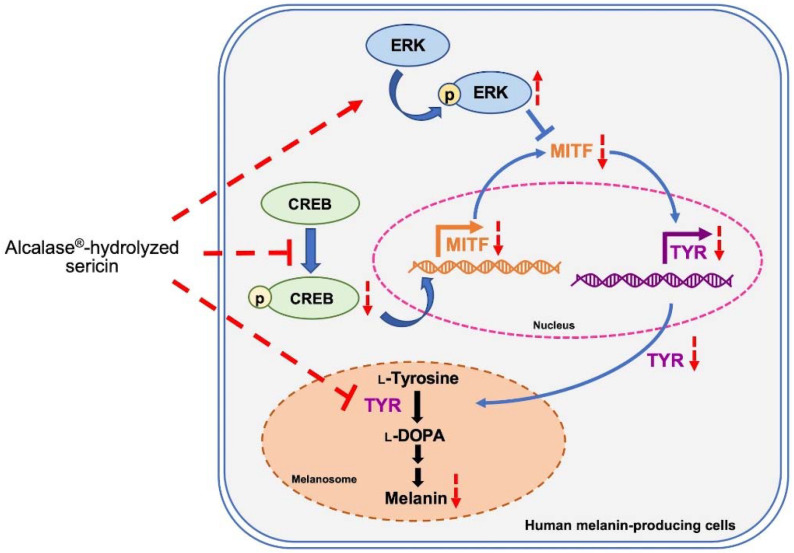
Alcalase^®^-hydrolyzed sericin suppress melanogenesis in human melanin-producing cells through modulating MITF expression as indicated with downregulated pCREB and overexpression of pERK. Not only the reduction of tyrosinase (TYR) protein level associated with the moderated MITF level, but sericin hydrolysates modified by Alcalase^®^ also directly inhibit the enzymatic function of human tyrosinase consequence with lowered melanin content. The red arrows and red symbols indicate the effect of sericin hydrolysates in human melanin-producing cells.

## Data Availability

The data presented in this study are available in Recycled sericin hydrolysates modified by Alcalase^®^ suppress melanogenesis in human melanin-producing cells via modulating MITF.

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
