# Peer review of "Recycled Sericin Hydrolysates Modified by Alcalase® Suppress Melanogenesis in Human Melanin-Producing Cells via Modulating MITF"

_ijms, 2022, doi:10.3390/ijms23073925_

Round 1
Reviewer 1 Report
An interesting original study proposing in vitro the use of sericin hydrolysates in vitro in the treatment of hyperpigmentation. I wasn't aware of the existence of this molecule, and further in vivo studies will have to be performed before considering this molecule as a possible treatment for melasma and other hyperpigmentations; Still, I found the paper very interesting, and eligible to be published after minor revisions:
line 43 you should add: in other to manage this condition also laser and light sources have been proposed with a good aesthetic outcome, but with a greater risk of side effects" and cite : https://doi.org/10.3390/app11167478 and doi: 10.1089/photob.2020.4850.
4.10 statistical analysis.
The sentence in lines 439-44o seems to have been cut, with no residual sense....please modify.
Good Luck
Author Response
Response to reviewer
Reviewer 1
- An interesting original study proposing in vitro the use of sericin hydrolysates in vitro in the treatment of hyperpigmentation. I wasn't aware of the existence of this molecule, and further in vivo studies will have to be performed before considering this molecule as a possible treatment for melasma and other hyperpigmentations; Still, I found the paper very interesting, and eligible to be published after minor revisions:
Response: The valuable comments from reviewer are appreciated. The suggestion of further investigation in in vivo study was added in the Discussion section (line 273-275) of revised manuscript as “Nevertheless, the anti-melanogenic effect of Alcalase®-hydrolyzed sericin should be further investigated in animal and clinical studies.”
- line 43 you should add: in other to manage this condition also laser and light sources have been proposed with a good aesthetic outcome, but with a greater risk of side effects" and cite : https://doi.org/10.3390/app11167478 and doi: 10.1089/photob.2020.4850.
Response: The Introduction section (line 47-49) of revised manuscript had been corrected as “In other to manage this condition also laser and light sources have been proposed with a good aesthetic outcome, but with a greater risk of side effects [6, 7].
References as presented in the manuscript:
- Nisticò, S.P.; Tolone, M.; Zingoni, T.; Tamburi, F.; Scali, E.; Bennardo, L.; Cannarozzo, G. A New 675 nm laser device in the treatment of melasma: Results of a prospective observational study. Photobiomodul. Photomed. Laser. Surg. 2020, 38, 560-564. Doi: 10.1089/photob.2020.4850.
- Silvestri, M.; Bennardo, L.; Zappia, E.; Tamburi, F.; Cameli, N.; Cannarozzo, G.; Nisticò, S.P. Q-switched 1064/532 nm laser with picosecond pulse to treat benign hyperpigmentations: A single-center retrospective study. Appl. Sci. 2021, 11, 7478. Doi: 10.3390/app11167478.
4.10 statistical analysis. The sentence in lines 439-44o seems to have been cut, with no residual sense....please modify.
Response: The sentence was modified in the Materials and Methods section (line 446-447) of revised manuscript as following “SPSS version 22 (IBM Corp., Armonk, NY, USA) with one-way analysis of variance (ANOVA) followed by Tukey’s post-hoc test was performed for statistical analysis.”
Minor correction
Grammar and typing error had been proofed and corrected by an English native speaker
*All changes in the revised manuscript are marked with red color.

Reviewer 2 Report
An interesting in vitro study proving the effectiveness of recycled sericin hydrolysates modified by Alcalase in the reduction of melanogenesis, proposing the compound as a possible treatment for all skin-related hyperpigmentation; only minor queries before publication.
Although you state that other depigmenting agents have side effects, in the introduction, you do not state clearly the current situation and products used for depigmenting; Please add a paragraph accordingly.
There are some trunked sentences in the text.....please revise
Thank You
Author Response
Response to reviewer
Reviewer 2
An interesting in vitro study proving the effectiveness of recycled sericin hydrolysates modified by Alcalase in the reduction of melanogenesis, proposing the compound as a possible treatment for all skin-related hyperpigmentation; only minor queries before publication.
- Although you state that other depigmenting agents have side effects, in the introduction, you do not state clearly the current situation and products used for depigmenting; Please add a paragraph accordingly.
Response: The Introduction section (line 43-49) of revised manuscript had been corrected as “Though various depigmenting agents such as kojic acid, arbutin and glycolic acid have been introduced for hyperpigmentation treatment instead of hydroquinone, a first line therapy which exhibits blue-black pigmentation after a long-term application, most of them possess a short span of efficacy and skin irritation [4, 5]. In other to manage this condition also laser and light sources have been proposed with a good aesthetic outcome, but with a greater risk of side effects [6, 7].
References as presented in the manuscript:
- Parvez, S.; Kang, M.; Chung, H.S.; Cho, C.; Hong, M.C.; Shin, M.K.; Bae, H. Survey and mechanism of skin depigmenting and lightening agents. Phytother. Res. 2006, 20, 921-934. Doi: 10.1002/ptr.1954.
- García-Gavín, J.; González-Vilas, D.; Fernández-Redondo, V.; Toribio, J. Pigmented contact dermatitis due to kojic acid. A paradoxical side effect of a skin lightener. Contact. Dermatitis. 2010, 62, 63-64. Doi: 10.1111/j.1600-0536.2009.01673.x.
- Nisticò, S.P.; Tolone, M.; Zingoni, T.; Tamburi, F.; Scali, E.; Bennardo, L.; Cannarozzo, G. A New 675 nm laser device in the treatment of melasma: Results of a prospective observational study. Photobiomodul. Photomed. Laser. Surg. 2020, 38, 560-564. Doi: 10.1089/photob.2020.4850.
- Silvestri, M.; Bennardo, L.; Zappia, E.; Tamburi, F.; Cameli, N.; Cannarozzo, G.; Nisticò, S.P. Q-switched 1064/532 nm laser with picosecond pulse to treat benign hyperpigmentations: A single-center retrospective study. Appl. Sci. 2021, 11, 7478. Doi: 10.3390/app11167478.
- There are some trunked sentences in the text.....please revise
Response: All grammar and typing errors had been proofed and corrected by an English native speaker
*All changes in the revised manuscript are marked with red color.
